# In the COVID-19 Era, Effects of Job Stress, Coping Strategies, Meaning in Life and Resilience on Psychological Well-Being of Women Workers in the Service Sector

**DOI:** 10.3390/ijerph19169824

**Published:** 2022-08-09

**Authors:** Hee-Kyung Kim

**Affiliations:** Department of Nursing, Kongju National University, Gongju 32588, Korea; hkkim@kongju.ac.kr; Tel.: +82-41-850-0304

**Keywords:** meaning of life, job stress, coping strategy, resilience, well-being, women, COVID-19

## Abstract

The purpose of this study is to analyze the factors affecting the psychological well-being by using variables of job stress, coping strategies, meaning of life, and resilience to improve the quality of working life during COVID-19. The subjects were 135 adult women working for banks. Data were collected by having the subjects fill out a paper-and-pencil questionnaire, and analyzed through *t*-test, ANOVA, Pearson’s correlation coefficients, and multiple regression analysis. The subjects’ psychological well-being showed positive correlations with the social support-seeking coping mechanism (r = 0.33, *p* < 0.001), problem-solving-focused coping mechanism (r = 0.55, *p* < 0.001), meaning in life (r = 0.45, *p* < 0.001), and resilience (r = 0.37, *p* < 0.001). Psychological well-being showed negative correlations with job stress (r = −0.44, *p* < 0.001) and avoidance-focused coping mechanism (r = −0.28, *p =* 0.001). The factors affecting the psychological well-being were problem-solving-focused coping mechanism (β = 0.35, *p* < 0.001), job role stress (β = −0.24, *p* < 0.001), meaning inlife (β = 0.29, *p* < 0.001), avoidance-focused coping mechanism (β = −0.23, *p* < 0.001), and resilience (β = 0.15, *p* = 0.023). It is necessary to formalize psychological intervention to induce the improvement of the quality of work life by increasing the psychological well-being of working women during the COVID-19 pandemic. It is suggested that intervention is made in consideration of variables identified as influencing factors to increase the psychological well-being of women workers.

## 1. Introduction

The COVID-19 disease, first reported in China in December 2019 is a new respiratory infection. The implementation of unprecedented strict quarantine measures in China has kept a large number of people in isolation and affected many aspects of people’s lives. It has also triggered a wide variety of psychological problems, such as panic disorder, anxiety, and depression [1]. Many countries have not been able to respond systematically, appropriately, and exemplarily to the sudden outbreak of COVID-19 around the world. The unbelievably poor response has left the people in chaos and left many dead. In order to prevent transmission and spread, the Korean government and local governments systematically and exemplarily responded through active infection control, social distancing, vaccination, and intensive treatment [2]. However, the prolonged COVID-19 pandemic caused the rapid changes in the overall society, such as personal welfare, social inequality, health problem, and bipolarization problems. Because of this, the public is experiencing various psychological problems, such as depression, stress, anxiety, and anger, which eventually lowers quality of life in general. Additionally, many people are psychologically unstable, becoming easily stressed by small things, cannot fully control their anger, and blame others for their problems [3]. In particular, after the COVID-19, in Korean society, the burden of care was further expanded on women, increasing the stress of working women and lowering job satisfaction [4]. Additionally, for employed women or single parents, women are disproportionately responsible for the bulk of domestic tasks, including childcare and eldercare in the USA [5]. In China, in the early days of the COVID-19 pandemic, many people showed symptoms of depression, anxiety, and stress, and psychological shocks were even greater in vulnerable individuals, such as women and people with poor health [6]. Service workers face, communicate, and provide professional services for the general population when performing their duties. Therefore, the risk of infection is relatively higher compared to other occupations, and stress levels related to the job, physical tension, and fatigue are also high [2]. This situation suggests that psychological interventions should be formulated to improve the mental health of vulnerable subjects during the COVID-19 pandemic.

As the industrial structure is changed in the modern society of Korea, a large proportion of women who participate in economic activities are working for the service industry [7]. Service workers respond directly to customers to address consumer needs, and they are exposed a multitude of stresses related to work in relation to customers, so they are experiencing psychological stresses [7]. Moreover, regardless of how they feel, they sometimes suffer from mental health problems or stress in the process of responding to customers’ demands [8]. It would be necessary to research only women as the vulnerable subjects by separating women from men. Therefore, it would be necessary for healthcare specialists, including nurses, to pay attention to the health of women workers in the service sector. In particular, a representative worker among women service workers can be a banker. The women workers try hard to achieve psychological well-being and life satisfaction by keeping a balance between work and family and controlling their lives for themselves [9], so it is necessary to research the psychological well-being of women office workers. Consequently, it would be necessary to provide healthcare interventions to increase mental health well-being, targeting women bank clerks who directly respond to customers and provide services for them. 

Meanwhile, the quality of working life (QWL) of office workers has recently emerged as a very important concept. QWL refers to employees’ efforts to improve working conditions, job, work safety, security, benefits, and compensation [10]. The main focus of the QWL is that the work makes employees’ lives better [11]. Therefore, employers in the service industry should strive to improve the quality of life of women employees. In addition, organizations must provide the resources that employees need to improve their work-related quality of life [12]. In order to improve the job-related QWL of working women, job satisfaction, job stress, financial compensation, work-life balance, working conditions, and job environment factors should be considered [11]. First of all, it is necessary to create a psychologically-stable state while increasing satisfaction and lowering stress for working women. Therefore, it is necessary to analyze the factors affecting psychological well-being.

Psychological well-being means all the psychological aspects that are regarded as composing the quality of an individual’s life [13]. Psychological well-being increases life satisfaction and positive effects and also decreases negative effects [14], so there should be efforts to enhance the psychological well-being of women service workers who are experiencing job-related stresses. The factors affecting the psychological well-being could be explained through various factors, including the mental, psychological, and social factors at a personal and organizational level. 

First, job stress and coping strategies were considered as variables at the organizational level. Job stress is the stress generated during the process of job performance. Because of unsafe working conditions/environments, job content, heavy workload, scarcity of work, interpersonal conflicts at work, role ambiguity, and job insecurity, members of workforce can feel uncomfortable, pressured, tense, and conflicted [15], which must be managed. Additionally, as the measures for managing the job stress, stress coping strategies could be considered to achieve well-being by successfully adjusting oneself within job-related stress [16], so it would be necessary to understand the degree of coping strategies of women service workers and also to use them as the basic data for healthcare interventions. Of all the cognitive/behavioral methods to handle the internal/external demands that threaten the capacity of individuals, coping strategies are a core mechanism that mediate stressful events and adjustment [17]. Coping strategies need to be examined for the improvement of perception and positive solutions in stressful situations. As coping strategies are provided through the process of socialization, beliefs or values of individuals are formed and the perception of stressful situations and the selection of a given strategy is influenced. Thus, this study aims to concretely analyze coping strategies by dividing them into social support-seeking coping strategies, problem-solving-focused coping strategies and avoidance-focused coping strategies [16]. This study aims to verify which coping strategy is used and the effects of those coping strategies on psychological well-being when the women service workers face stressful situations and then utilize the strategies to increase the psychological well-being afterwards. 

Next, as variables that could play mediating roles for overcoming job stress and achieving the psychological well-being of women workers in the service sector, meaning in life and resilience should be considered. Meaning in life could be considered as a psychological element. Meaning in life provides a reason for humans to live and is the source of happiness. The meaning of life is related to purpose and importance [18]. Overall, people with a high sense of meaning in their lives show high psychological well-being, tend to be sincere and trust others, and show a low tendency towards depression, anxiety, and neurosis [18,19]. As an important factor of mental health [18], the meaning of life is an important element affecting happiness and also an index of psychological health related to happiness [20]. It also helps psychological adjustment even in stressful situations [18]. Thus, it could be inferred that women workers in the service sector who work harder to find meaning in their life during the hardship and difficulties of the COVID-19 era could achieve psychological well-being by finding the energy for their growth and development. Lastly, as the women service workers are placed in various stressful situations, resilience that helps recover and positively change life patterns is an element that provides and has positive effects on psychological stability [21]. People with resilience can flexibly respond to stressful situations and have the dynamic and adaptive ability to maintain or improve balance when facing changing environments [22]. It also reduces the influence of risk factors, such as stress on mental health. People with high resilience can show a quick recovery by re-evaluating stress, raising their resilience to stress, controlling negative emotions, and utilizing positive effects [21]. Thus, resilience can become an ability for women service workers to cope with various stresses, with work as a mediating variable even in the process of discovering the meaning of life, and become an important factor affecting well-being [23]. Therefore, to increase the well-being of women service workers, resilience was included as a variable. 

Based on the contents above, this study aims to uncover the degree of the meaning of life, job stress, coping strategies, resilience, and psychological well-being of women service workers in order to analyze their relations, to verify the factors affecting the psychological well-being, and to utilize them as basic data for seeking healthcare intervention measures for the psychological well-being of working women who respond to customers and provide services in the COVID-19 era. This research design is shown in Figure 1.

## 2. Materials and Methods

### 2.1. Participants

The research participants are a total of 135 adult women ranging from 21–51 years old who perform customer service while working for 20 branches of H and N banks in C_1_ and C_2_ cities in C province, G city, D city, and S city. Individuals included were those who understood the purpose of this study, gave written consent by voluntarily revealing their intention to participate, and they had been working for six months or more. The exclusion criteria were men, those with less than 6 months of work experience, or those in charge of administrative affairs. For the calculation of the number of samples in this study, the method of calculating the number of samples in previous studies [24] for the regression analysis was used. In the process of analyzing the number of subjects required in this study by using the G-power program 3 [25], a total of 130 samples was needed to maintain a total of seven predictor variables, 0.15 effect size, 0.05 significance level, and 0.90 test power. The seven variables were meaning of life, job stress, social support-seeking coping strategies, problem-solving-focused coping strategies, avoidance-focused coping strategies, resilience, and economic status. Considering the 10% drop-out rate, the survey was conducted for 143 people. A total of eight questionnaires with omitted or inappropriate responses were removed and 135 questionnaires were ultimately used.

### 2.2. Procedures

The data collection procedures were as follows: 1. First it was approved by the Institutional Review Board of the university to proceed with this study. 2. The researcher visited two banks, namely H bank and N bank in G city of C province, explained the purpose and methods of this study to the branch managers, and then obtained permission for data collection. 3. The branch managers who allowed the data collection explained the research contents in the executive meeting with other branches, and the branch managers who were willing to cooperate with this study took as many questionnaires as the number of women bank clerks. 4. The women bank clerks read the explanation of this study and then signed a written consent form. After that, the women bank clerks filled out total seven pages of the questionnaire to measure the job stress, coping strategies, meaning of life, resilience, psychological well-being, and general characteristics. The completed questionnaires were sent to the branch initially contacted through the bank network. 5. The researcher collected the completed questionnaires. It took about 15 min for the subjects to complete the questionnaire. From April to May 2022, the data were collected from a total of 20 bank branches of H bank and N bank in C1 and C2 cities of S city, D city, and G city of C province. 

### 2.3. Measures

The tool explanation is as follows. The meaning of life, job stress, three coping strategies, and resilience are independent variables. Psychological well-being is a dependent variable. Religion included in the general characteristics may be a potential confounding variable. The definitions of each variable were added.

#### 2.3.1. Job Stress

Job Stress is a physical and psychological negative reaction felt by interacting with the job environment, that is, stimulation in performing the job [26]. This study used the modified job stress tool focusing on elements revealed in the research by Lee [27] and Beehr and Newman [28]. This tool has total of twenty items, including subareas, such as six items concerning role (role ambiguity, role conflict, and role overload), five items about interpersonal relationship (frustration, relationship with supervisor, relationship with colleagues, and relationship with subordinates), five items about environment (indirect environment and direct environment), and four items about interaction (participation in decision-making, the limitation of organizational structure, and the collectivity of personnel policy). Each item is based on a 5-point rating scale. The average score ranges from 1 to 5. (1, Not at all; 2, Not really; 3, No; 4, A little bit; 5, Very much so). The higher average score, the higher the job stress. In the research by Lee [26], the reliability Cronbach’s α was 0.84, including the subareas, such as role factors (0.70), interpersonal relationships (0.80), environment (0.77), and interactions (0.85). In this study, it was shown to be 0.91, including subareas such as role factors (0.61), interpersonal relationships (0.95), environment (0.86), and interactions (0.88).

#### 2.3.2. Coping Strategies

Coping Strategies are the cognitive and behavioral efforts used to handle the internal/external demands that could threaten the capacity of individuals [17]. This study used the Coping Strategy Indicator developed by Shin and Kim [29]. Composed of a total of 33 items, this tool has total three subareas: social support-seeking coping strategies, problem-solving-focused coping strategies, and avoidance-focused coping strategies. Each item is based on a 3-point rating scale (1, Not at all; 2, A little bit; 3, Very much so). It means that the higher the average score in each subarea of social support-seeking, problem-solving-focused, and avoidance-focused responses, the more each response is used. When the tool was initially developed, the reliability Cronbach’s α of the social support-seeking coping mechanism was 0.90; the problem-solving-focused coping mechanism was 0.88; and the avoidance-focused coping mechanism was 0.67. In this study, the subareas were shown to be social support-seeking coping strategies (0.88), problem-solving-focused coping strategies (0.87), and avoidance-focused coping strategies (0.72).

#### 2.3.3. Meaning in Life

Meaning in life is a subjective feeling that one’s life is meaningful, includes the process of discovering, creating, and searching for meaning [30]. As something that should be endlessly pursued and realized by continuous efforts in the process of everyday life, this study used the Korean-version Meaning in Life Questionnaire (MLQ) developed by Won et al. [18] and Steger et al. [30] and then adapted and validated by Won et al. [18]. It has a total of ten items, including five items about the pursuit of meaning and five items about the discovery of meaning. Each item is based on a 7-point rating scale (1, Not at all; 2, Generally not like that; 3, Not like that a little bit; 4, Moderate; 5, A little bit; 6, Mostly like that; 7, Very much so). The average scores range from 1 to 7. A higher average score means an individual realize the meaning in life more. The reliability Cronbach’s α of the tool in the research by Won et al. [18] was 0.88, whereas in this study, it was 0.90, including the pursuit of meaning (0.88) and discovery of meaning (0.78).

#### 2.3.4. Resilience

Resilience is the ability to adapt successfully by responding flexibly based on changing situational needs or appropriate self-control in a stressful environment [31]. This study used the resilience measurement tool developed by Block and Kremen [31] and then adapted by Yoo and Shim [32]. Composed of a total of 14 items, each item is based on a 4-point rating scale (1, Not at all; 2, Sometimes not like that; 3, Sometimes like that; 4, Very much so). The average score ranges from 1 to 4. The higher the average score, the higher the degree of resilience. In the research by Yoo and Shim [32], the reliability Cronbach’s α was 0.76, while it was shown to be 0.77 in this study. 

#### 2.3.5. Psychological Well-Being

Psychological Well-Being is an emotion that accompanies actions expressed in the process of realizing one’s potential [13]. In this study, a tool developed by Ryff [13] and modified by Kim et al. [14] and used by Song [33] was utilized. This tool is composed of a total of 44 items, including a total of six subareas, such as eight items about self-acceptance, seven items about positive interpersonal relationships, seven items about autonomy, seven items about environmental control, seven items about the purpose of life, and eight items about personal growth. Each item is based on a 5-point rating scale (1, Not at all; 2, No; 3, Moderate; 4, Yes; 5, Very much so). The average scores range from 1 to 5. 

The higher the average score, the higher the psychological well-being. In the research by Song [33], the reliability Cronbach’s α was 0.91, including the subareas, such as self-acceptance (0.77), positive interpersonal relationships (0.79), autonomy (0.69), environmental control (0.66), purpose of life (0.76), and personal growth (0.64). In this study, the total reliability was 0.89, including the subareas, such as self-acceptance (0.70), positive interpersonal relationships (0.74), autonomy (0.41), environmental control (0.65), purpose of life (0.75), and personal growth (0.63). 

### 2.4. Statistical Analyses

Using the IBM SPSS Window 25.0 Program [34], the collected data were data-processed, and the methods of data analysis are as follows:For the degree of general characteristics, the meaning of life, job stress, coping strategies, resilience, and the psychological well-being of subjects, descriptive statistics, such as frequency, percentage, mean, and standard deviation, were used.For differences in the psychological well-being of subjects, according to the general characteristics, this study used the *t*-test and ANOVA, and Scheffé’s test as post-test.The correlations of meaning of life, job stress, coping strategies, resilience, and psychological well-being of subjects were analyzed through the Pearson’s correlation coefficient.To examine the factors affecting the psychological well-being of subjects, this study used the stepwise multiple regression analysis after diagnosing the multicollinearity.

### 2.5. Ethical Principles

This study was approved by the K University’s Institutional Review Board for the purpose, methodology, and protection of the rights of participants (KNU_IRB_2022-34). After explaining the purpose, background, methods, and procedure of this study, possible risks and benefits to research participants, the matter of discontinuing the participation in this study, and the confidentiality of personal information to the individuals with the intention to participate, they were told to submit their written consent, and their autonomous participation was respected. They were also informed that the collected data would be stored for three years in a cabinet with safety device, that the encoded results would be stored/managed in a computer, and that the data would be discarded by using a shredder three years after being stored.

## 3. Results

### 3.1. General Characteristics of Subjects

The general characteristics of subjects are as follows. The total number of women service workers was 135. The average age was 38.92 ± 6.89 years, the range was 21~51 years, and 68 persons (50.4%) were aged 40–51 years. In terms of marital status, there were 115 married individuals (85.2%). There were 82 people (60.7%) with no religion. Regarding education, there were 126 participants (93.3%) who had graduated from college, university, or graduate school. The mean length of working experience was 15.15 ± 7.41 years. People with working experience of 12 years or more numbered 108 (80.0%). Subjects who had no experience of receiving education regarding psychological well-being for at least one year accounted for 125 people (92.6%). The subjects with a monthly income of less than 4 million KRW were 69 individuals (51.1%). Subjects with moderate economic status numbered 115 people (85.2%). Regarding health status, 125 subjects who evaluated themselves as healthy (92.6%). In terms of sleeping status, subjects who slept well comprised 122 people (90.4%) (Table 1).

### 3.2. Differences in Psychological Well-Being According to the General Characteristics of Subjects

The differences in psychological well-being according to the general characteristics of women service workers are as follows: In the economic status of subjects, the subjects who responded as “high” showed a higher degree of psychological well-being at the statistically-significant level than the subjects who responded as “low–moderate” (t = 2.21, *p* = 0.029) (Table 1).

### 3.3. Degree of Job Stress, Coping Strategies, Meaning in Life, Resilience, and Psychological Well-Being of Subjects

Job stress was 2.63 ± 0.60 points out of 5, and subareas were shown as stress about job role at 2.78 ± 0.55 points, stress about interpersonal relationships at 2.28 ± 0.7 points, stress about job environment at 2.39 ± 0.80 points, and the highest stress was concerned with interactions at 3.14 ± 0.92 points. Regarding coping strategies, the social support-seeking coping mechanism was 2.22 ± 0.40 points out of 3; the problem-solving-focused coping mechanism was 2.29 ± 0.34 points; and the avoidance-focused coping mechanism was 1.74 ± 0.33 points. The degree of meaning of life of women service workers was 5.32 ± 0.92 points out of 7 and, among the subareas, the pursuit of meaning was 5.43 ± 0.97 points and the discovery of meaning was 5.22 ± 0.94 points. Resilience was 2.76 ± 0.41 points out of 4. Psychological well-being was 3.48 ± 0.39 points out of 5, including subareas such as the degree of self-acceptance at 3.47 ± 0.45 points, degree of positive interpersonal relationships at 3.48 ± 0.66 points, degree of autonomy at 3.31 ± 0.40 points, degree of environmental control at 3.55 ± 0.47 points, degree of the purpose of life at 3.66 ± 0.53 points, and degree of personal growth at 3.45 ± 0.45 points (Table 2). 

### 3.4. Relations of Job Stress, Coping Strategies, Meaning in Life, Resilience, and Psychological Well-Being of Subjects

The psychological well-being of subjects showed positive correlations with social support-seeking coping mechanism (r = 0.33, *p* < 0.001), problem-solving-focused coping mechanism (r = 0.55, *p* <0.001), the discovery of meaning of life (r = 0.44, *p* < 0.001), pursuit of meaning of life (r = 0.42, *p* < 0.001), meaning in life (r = 0.46, *p* < 0.001), and resilience (r = 0.37, *p* < 0.001). On the other hand, it showed negative correlations with job role stress (r = −0.43, *p* < 0.001), interpersonal stress (r = −0.41, *p* < 0.001), job environment stress (r = −0.40, *p* < 0.001), interactional stress (r = −0.20, *p* = 0.019), job stress (r = −0.44, *p* < 0.001), and avoidance-focused coping mechanism (r = −0.28, *p* = 0.001) (Table 3). 

### 3.5. Factors Affecting the Subjects’ Psychological Well-Being

To verify the relative influences of the factors that could explain the psychological well-being of women service workers, economic status, which showed significant differences in psychological well-being according to the general characteristics, was added as a variable and analyzed by changing it to a Dummy variable. Additionally, the stepwise multiple regression analysis was conducted by inserting independent variables, such as the meaning of life, the discovery of meaning, the pursuit of meaning, job stress, job role stress, interpersonal stress, job environment stress, interactional stress, social support-seeking coping strategies, problem-solving-focused coping strategies, avoidance-focused coping strategies, and resilience. 

In the results of examining the plot using the homogeneity of variance test, the homoscedasticity was verified, and the value of Durbin–Watson for verifying the independence of the residual was shown as 2.02, which satisfied the hypothesis of independence. In the results of examining the P-P chart for verifying the independence for verifying the normality of error term, the normal distribution was shown. Additionally, in the evaluation of multicollinearity between independent variables, the tolerance limit was 0.83~0.91, which was higher than 0.1. The Variance Inflation Factor (VIF) of each variable was 1.10~1.21, which was not over 10. Thus, it satisfied the basic hypothesis of homoscedasticity and regular distribution of residual. 

In the results of verifying the relative influences of the factors affecting the psychological well-being of women service workers in the COVID-19 era, the regression model was statistically significant (F = 30.46, *p* < 0.001). The factors affecting the psychological well-being were problem-solving-focused coping mechanism (*β* = 0.35, *p* < 0.001), job role stress (*β* = −0.24, *p* < 0.001), meaning in life (*β* = 0.29, *p* < 0.001), avoidance-focused coping mechanism (*β* = −0.23, *p* < 0.001) and resilience (*β* = 0.15, *p* = 0.023). The explanatory power of those five variables was 52.4%, and the variable with the biggest influence among them was problem-solving-focused coping mechanism (Table 4). 

## 4. Discussion

This study analyzed the factors affecting the psychological well-being of women workers in the service sector during the COVID-19 era, provided basic data for the development of psychological well-being intervention programs, and ultimately provided clues to improving the quality of work life. 

In the COVID-19 era, the job stress of women workers in the service sector was 2.63 points out of 5, and the stress caused by job role was the highest (2.78). The degree of psychological well-being was 3.48 out of 5, so there should be the measures for lowering stress and improving psychological well-being. Job stress is one of the most common health problems, especially for women, in many organizations [35].

In the research by Almeida et al. [5], the COVID-19 pandemic caused stress for everyone, especially for women, and because working women have the high burden of handling both housework and job-related tasks, stress management measures should be established for women. Additionally, during the COVID-19 era, it was reported that Chinese women also showed a higher level of stress, anxiety, depression, and post-traumatic stress symptoms; serious psychological/mental results in general; and significantly higher stresses when they were employed [6]. Additionally, in research targeting Italian university students during the COVID-19 era by Giusti et al. [36], university students experienced post-traumatic symptoms, such as stress, depression, psychological pain, and anxiety, and the students who belonged to vulnerable groups were required to undergo psychological intervention for the improvement of mental health.

In addition, Sriharan et al. [37] reported that women are highly likely to develop mental health problems, such as stress, fatigue, and depression, as well as personal, organizational, and social problems, including a lack of social support, excessive workload, and stress, according to a study of women working in healthcare during the COVID-19 pandemic. Additionally, it was said that the problem of the system could act as a trigger for negative consequences. Looking at the above, it was found that women working in various countries during the COVID-19 had stress related to life and job. Therefore, gender inequality may occur in the COVID-19 era, so gender should be considered, and programs should be established to relieve stress, improve mental health, and improve psychological well-being. 

During the COVID-19 era, the psychological well-being of women service workers showed positive correlations with the overall meaning of life and its subareas, such as the pursuit of the meaning of life and discovery of the meaning in life, social support-seeking copingmechanism, problem-solving-focused coping mechanism, and resilience, while it showed negative correlations with overall job stress and subareas such as job role stress, interpersonal stress, job environment stress, and interactional stress, and avoidance-focused copingmechanism. The factors affecting the psychological well-being of women service workers were problem-solving-focused copingmechanism, job role stress, the discovery of meaning in life, avoidance-focused copingmechanism, and resilience; and the variable with the greatest effects of these during the COVID-19 era was problem-solving-focused copingmechanism. Job stress, such as work ambiguity and the conflict of roles for women workers reduced their satisfaction and psychological well-being [38]. Job role stress caused negative results in the mental health of women office workers [39,40]. Job stress aggravated by providing their services in person during the COVID-19 era could have had the greatest effects on anxiety [41], which could have led to negative results of psychological well-being [2]. In the results of a study targeting men and women workers in 35 European countries by Mensah [35], job stress had negative effects on the psychological well-being of women workers and social support played a mediating role. Thus, the support from colleagues and supervisors at work is important, and the variable of social support should be considered for reducing work stress and improving psychological well-being. 

Additionally, job stress, problem-solving-focused coping strategies mechanism, and emotion-focused coping mechanism were major factors affecting the psychological well-being of office workers [26], which supports the results of this study. In relation to jobs, role ambiguity, role conflict, and role overload are factors of job stress [27], so psychological well-being should be increased by raising psychological well-being through a concrete analysis of job stress factors and also applying intervention programs, such as mindfulness [41]. In the results of the research on psychological well-being targeting the Korean university students by Yoon [42], the problem-solving-focused coping strategies had indirect effects on psychological well-being through the meaning of life. Additionally, the emotion-focused coping strategies, including avoidance-focused coping strategies, had direct effects on it. In a model with the mediation of spiritual meaning, the problem-solving-focused coping mechanismhad direct effects on psychological well-being, and the emotion-focused coping mechanism showed indirect effects on it through spiritual meaning. In the study of abused women by Rodriguez [43], women used coping strategies for overcoming the shock of incidents and pain, which was an important element for achieving psychological well-being: when their psychological well-being was high, crises could be overcome through problem-solving-focused coping mechanism. When their psychological well-being was low, they used emotion-focused coping mechanism. Thus, in the COVID-19 era, there should be some programs that help the women service workers to achieve psychological well-being by encouraging them to use a suitable coping strategy for themselves, such as problem-solving-focused coping or avoidance-focused coping mechanism. 

Additionally, meaning in life and resilience were major factors affecting the psychological well-being of women service workers. The results of the research by Chang [20] targeting middle-aged women during the COVID-19 era, demonstrated that meaning in life showed a positive correlation with well-being. In the study targeting a total of 179 mothers of a mean age of 29 years [19], meaning of life showed a strong positive correlation with subjective well-being. The meaning of life in university students also showed high correlation with psychological well-being [42], and in the age group of the subjects of this study, pursuing and discovering the meaning of life was closely related to psychological well-being, which supports the results of this study. As an important factor for mental health [18], the meaning in life is an index of psychological health related to happiness as a factor significantly affecting well-being [20]. Additionally, it helps psychological adjustment even in stressful situations [18]. Therefore, women service workers who try harder to find the meaning in life can achieve psychological well-being by obtaining the energy for growth and development during hardship and difficulties of the COVID-19 era, so the leaders at work need to provide opportunities to find meaning in life through after-work programs. Additionally, psychological well-being showed positive correlations with subareas of meaning in life, such as the pursuit of meaning and the discovery of meaning. The pursuit of the meaning of life is a motive for finding meaning in one’s own life, while the discovery of meaning is to subjectively feel that one’s own life is meaningful [44]. In order for the pursuit of meaning to be continued to the discovery of meaning, there should be an intervention that helps individuals to realize and introspect their personal meaning [45].

Next, the resilience was a further contributing factor. In the COVID-19 era, the resilience of early childhood teachers was a factor affecting their psychological well-being [46], and in the same period, the well-being of middle-aged women had effects on resilience [23], which was similar to the results of this study. Due to the unexpected COVID-19 pandemic, people faced various difficulties [3]. Resilience has positive effects on mental health, helping individuals to view their lives as optimistic and positive. It also helps job performance, which leads to the reduction of stress [46]. Thus, to increase the psychological well-being of women service workers, it would be necessary to support various resources for raising resilience even in stressful situations. Therefore, it is required to utilize intervention programs so that resilience can play a mediating role in improving psychological well-being and the quality of work life while adapting to the COVID-19 era.

Meanwhile, in research targeting the survivors from disasters by Biankini et al. [47], survivors showed positive or negative results after the shock of events and gender, especially, was a risk factor of post-traumatic growth, where women showed the moderate symptoms of depression. The mental health of women is regarded as more vulnerable to situations such as the COVID-19 pandemic. Paradoxically, however, crisis-related stress can bring about post-traumatic growth. Therefore, individuals who experience the shock of an event perceive the meaning of life and experience growth by changing hardships to something positive by fully utilizing stress coping strategies and increasing their resilience [20,48]. Moreover, in research targeting firefighters [49], the threat of stress and the evaluation of a challenge could mediate the relationship between resilience and post-traumatic growth, so it would be desirable to consider this result. Thus, in order for women service workers to achieve positive results of post-traumatic growth after the COVID-19 pandemic, it would be necessary to systematically support the application of stress management methods, the utilization of coping strategies through stress evaluation, and the development of resilience enhancement programs [50].

Finally, it is suggested that the results of this study are used as follows in terms of healthcare research, education, and practice. In order to maintain psychological well-being due to the prolonged COVID-19 pandemic, this research can be used as basic data for ultimately enhancing psychological well-being by classifying job-related stress for women in customer-facing service jobs and subdividing coping strategies to solve it. In terms of education, the industry leaders can use this study as a basis for education to increase the psychological well-being of working women responding to customers during the prolonged situation of COVID-19. At a practical level, it is necessary to subdivide and investigate the meaning of life, job stress, coping strategies, and the resilience of working women in the service industry and to develop and apply interventions to promote psychological well-being based on communities. However, in this study, instead of recruiting women working in various service fields, the sampling was conveniently performed in only some areas, so caution should be used in regard to generalizing the research results. Therefore, in preparation for unexpected situations, such as COVID-19, it is proposed to diversify the selection of regions, industries, and subjects. 

## 5. Conclusions

In the COVID-19 era, the women service workers experienced many difficulties when responding to customers. In order to improve the psychological well-being of women service workers in the COVID-19 era, this study aimed to analyze the factors affecting the psychological well-being by using variables such as the meaning in life, stress, coping strategies, and resilience. Problem-solving-focused copingmechanism, job role stress, the discovery of meaning in life, avoidance-focused copingmechanism, and resilience were verified as variables affecting psychological well-being in the COVID-19 era. Thus, there should be interventions that could reduce their stress about job roles, improve problem-solving-focused coping abilities, and reduce avoidance-focused copingmechanism. Additionally, there should be health-care interventions that could help individuals find meaning in their lives and increase resilience. Such results could be used as the basic data for systematically and analytically verifying factors that could enhance the psychological well-being of women service workers, accumulating the base data of practical work, and suggesting various research directions. Additionally, leaders who decide on policies and systems in industry need to establish relevant policies and provide administrative/financial support, while the women service workers should attempt to apply measures to improve psychological well-being for themselves.

## Figures and Tables

**Figure 1 ijerph-19-09824-f001:**
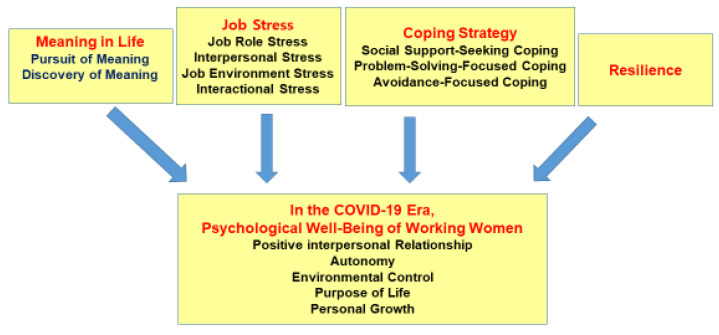
Research design of the study.

**Table 1 ijerph-19-09824-t001:** Differences in psychological well-being according to general characteristics (N = 135).

Variables	Division	Number	%	Mean	SD	t/F	*p*-Value
Age (year)	21–29	20	14.8	2.00	0.38	0.61	0.545
	30–39	47	34.8	2.30	0.52		
	40–51	68	50.4	2.27	0.58		
Marital status	Married	115	85.2	3.49	0.39	0.40	0.693
	Unmarried, etc.	20	14.8	3.45	0.43		
Religion	Yes	53	39.3	3.49	0.41	0.19	0.853
	No	82	60.7	3.47	0.38		
Education	Graduated from high school	9	6.7	3.63	0.39	1.17	0.246
	Above college	126	93.3	3.47	0.39		
Working experience (year)	<12	27	20	3.55	0.34	1.04	0.302
	≥12	108	80	3.46	0.40		
Educational experience (number)	0	125	92.6	3.48	0.39	0.10	−0.924
	≥1	10	7.4	3.49	0.4		
Monthly income (10,000 KRW)	<400	69	51.1	3.49	0.34	−0.97	0.334
	≥400	66	48.9	3.52	0.44		
Economic status	High	20	14.8	3.66	0.41	2.21	0.029
	Low~moderate	115	85.2	3.45	0.38		
Health status	Healthy	125	92.6	3.50	0.40	0.61	0.542
	Unhealthy	10	7.4	3.41	0.28		
Sleeping status	Sleeps well	122	90.4	3.50	0.40	1.70	0.091
	Does not sleep well	13	9.6	3.31	0.26		

Note: SD = standard deviation.

**Table 2 ijerph-19-09824-t002:** The degree of the research variables (N = 135).

Variables	Categories	Mean	SD	Range
Job stress	Job role stress	2.78	0.55	1.17–4.17
	Interpersonal stress	2.28	0.75	1.00–4.00
	Job environment stress	2.39	0.80	1.00–4.60
	Interactional stress	3.14	0.92	1.00–5.00
	Total	2.63	0.6	1.20–3.95
Coping strategy	Social support-seeking coping mechanism	2.22	0.40	1.27–3.00
	Problem-solving-focused coping mechanism	2.29	0.34	1.45–3.00
	Avoidance-focused coping mechanism	1.74	0.33	1.00–2.73
Meaning in life	Pursuit of meaning	5.43	0.97	2.60–7.00
	Discovery of meaning	5.22	0.94	2.00–7.00
	Total	5.32	0.92	2.70–7.00
Resilience		2.76	0.41	1.86–3.79
Psychological well-being	Self-acceptance	3.47	0.45	2.50–4.63
	Positive interpersonal relationships	3.48	0.66	1.86–4.71
	Autonomy	3.31	0.40	2.14–4.57
	Environmental control	3.55	0.47	2.29–4.71
	Purpose of life	3.66	0.53	2.00–5.00
	Personal growth	3.45	0.45	2.25–4.50
	Total	3.48	0.39	2.20–4.45

Note: SD = standard deviation.

**Table 3 ijerph-19-09824-t003:** Correlations of the variables (N = 135).

Variables	Job Role Stress R (*p*)	Interpersonal Stress R (*p*)	Job Environment Stress R (*p*)	Interactional Stress R (*p*)	Job Stress R (*p*)	Social Support-Seeking Coping R (*p*)	Problem-Solving-Focused Coping r (*p*)	Avoidance-Focused Coping r (*p*)	Pursuit of Meaning R (*p*)	Discovery of Meaning R (*p*)	Meaning of Life R (*p*)	Resilience R (*p*)	Psychological Well-Being R (*p*)
Job role stress	1												
Interpersonal stress	0.63 (<0.001)	1											
Job environment stress	0.63 (<0.001)	0.69 (<0.001)	1										
Interactional stress	0.40 (<0.001)	0.30 (<0.001)	0.58 (<0.001)	1									
Job stress	0.81 (<0.001)	0.81 (<0.001)	0.90 (<0.001)	0.71 (<0.001)	1								
Social support-seeking coping mechanism	−0.13 (0.133)	−0.27 (0.002)	−0.26 (0.002)	−0.16 (0.072)	−0.26 (0.003)	1							
Problem-solving-focused coping mechanism	−0.21 (0.014)	−0.24 (0.006)	−0.37 (<0.001)	−0.20 (0.022)	−0.32 (<0.001)	0.54 (<0.001)	1						
Avoidance-focused coping mechanism	0.25 (0.004)	0.23 (0.008)	0.14 (0.114)	0.13 (0.130)	0.23 (<0.001)	−0.02 (0.783)	−0.07 (0.396)	1					
Pursuit of meaning	−0.16 (0.064)	−0.27 (0.002)	−0.30 (<0.001)	−0.14 (0.110)	−0.27 (0.001)	0.16 (0.072)	0.29 (0.001)	0.01 (0.887)	1				
Discovery of meaning	−0.11 (0.198)	−0.20 (0.021)	−0.20 (0.021)	−0.08 (0.367)	−0.18 (0.032)	0.12 (0.160)	0.24 (0.006)	0.11 (0.186)	0.85 (<0.001)	1			
Meaning in life	−0.141 (0.102)	−0.24 (0.005)	−0.26 (0.002)	−0.11 (0.193)	−0.24 (0.005)	0.14 (0.096)	0.28 (0.001)	0.07 (0.453)	0.96 (<0.001)	0.96 (<0.001)	1		
Resilience	−0.15 (0.077)	−0.08 (0.352)	−0.14 (0.109)	−0.10 (0.262)	−0.14 (0.095)	0.12 (0.180)	0.33 (<0.001)	0.09 (0.308)	0.28 (0.001)	0.28 (0.001)	0.29 (0.001)	1	
Psychological well-being	−0.43 (<0.001)	−0.41 (<0.001)	−0.40 (<0.001)	−0.20 (0.019)	−0.44 (<0.001)	0.33 (<0.001)	0.55 (<0.001)	−0.28 (0.001)	0.44 (<0.001)	0.42 (<0.001)	0.45 (<0.001)	0.37 (<0.001)	1

*p*-value (continuous variables: Pearson’s correlational coefficients).

**Table 4 ijerph-19-09824-t004:** Factors affecting the subjects’ psychological well-being.

Variables	B	SE	β	t	*p*
Constant	2.46	0.28		8.85	<0.001
Problem-solving-focused coping mechanism	0.40	0.07	0.35	5.40	<0.001
Job role stress	−0.17	0.05	−0.24	−3.71	<0.001
Meaning in life	0.12	0.03	0.29	4.46	<0.001
Avoidance-focused coping mechanism	−0.27	0.08	−0.23	−3.67	<0.001
Resilience	0.15	0.06	0.15	2.31	0.023

*p*-value (continuous variables: multiple regression analysis). Note: SE = standard error. R^2^ = 0.541; Adjusted R^2^ = 0.524; F = 30.46; *p* < 0.001.

## Data Availability

The data underlying this article will be shared upon reasonable request from the author.

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
