# Peer review of "In the COVID-19 Era, Effects of Job Stress, Coping Strategies, Meaning in Life and Resilience on Psychological Well-Being of Women Workers in the Service Sector"

_ijerph, 2022, doi:10.3390/ijerph19169824_

Round 1
Reviewer 1 Report
The Authors have significantly improved their work, but we would appreciate including further suggestions.
A couple of essential suggestions:
1) The word “female” should be changed to “women” (first of all, in the title of the work!!!)
2) Results
In these sections, the results, already reported in Tables, do not need to be repeated but briefly commented.
Author Response
Reviewer 1
Thank you very much for taking the precious time to review it so that it can be a good paper.
|
Point out |
revised |
|
A couple of essential suggestions: 1) The word “female” should be changed to “women” (first of all, in the title of the work!!!) |
I changed the word “female” to “women” |
|
2) Results In these sections, the results, already reported in Tables, do not need to be repeated but briefly commented. |
Based on the reviewer's comments, we've reduced the results as much as possible. |

Reviewer 2 Report
Dear authors,
Congratulations on your effort. Incorporating the previous suggestions greatly improved the manuscript, i.e., the abstract covers all the important parts of the paper, the method section is clear, the discussion is sounder, and the implications are coherent.
Therefore, my recommendation is to accept the article.
Best,
Author Response
Thank you very much for taking the precious time to review it so that it can be a good paper.
Thank you for accepting the revised parts. There is nothing specifically pointed out and there is nothing to modify, so I checks again and uploads the file. Thank you so much.
I upload a file that reflects the correction requests of other reviewers. Thanks once again.

Reviewer 3 Report
The article appears to have changes made given previous comments. These sections in red add to the piece.
I do find some of the abstract confusing and it will probably be confusing to an average reader looking quickly to gauge interest. I.e. "in consideration of the problem-solving-focused coping, job stress, avoid-ance-focused coping, meaning of life" - I do not understand what this means and it is ill-defined to be included in an abstract.
"Like all over the world... Korea systematically and exemplarily responded" - No, nations all over the world did not respond systematically and exemplarily - many did an incredibly poor job of responding. Though certain aspects of South Korea's response were indeed exemplary.
Around line 47. Face-to-face… - This is not a good sentence. Rephrase.
Line 133 , and anywhere else the term occurs – still not sure what you mean by meaning of life – I think you mean quality of life. This is never well-defined in this paper. Because it is such an integral part of the argument, including in your statistical analysis, it must be well-defined or the paper doesn’t make sense. Even the explanation in 2.3.3 does not do much to explain and actually just ends up making the issue more confused.
Line 149 – why remove people with less work experience? Why this threshold?
Line separation on table 2 is confusing.
Author Response
Thank you very much for taking the precious time to review it so that it can be a good paper.
|
Point out |
revised |
|
The article appears to have changes made given previous comments. These sections in red add to the piece.
|
Thank you so much. |
|
I do find some of the abstract confusing and it will probably be confusing to an average reader looking quickly to gauge interest. I.e. "in consideration of the problem-solving-focused coping, job stress, avoidance-focused coping, meaning of life" - I do not understand what this means and it is ill-defined to be included in an abstract. |
I revised the sentence of the suggestion in the abstract as follows.
-> It is suggested that intervention is made in consideration of variables identified as influencing factors to increase the psychological well-being of women workers. |
|
"Like all over the world... Korea systematically and exemplarily responded" - No, nations all over the world did not respond systematically and exemplarily - many did an incredibly poor job of responding. Though certain aspects of South Korea's response were indeed exemplary.
|
changed sentence as follows. Many countries have not been able to respond systematically, appropriately, and exemplary to the sudden outbreak of COVID-19 around the world. The unbelievably poor response has left the people in chaos and left many dead. |
|
Around line 47. Face-to-face… - This is not a good sentence. Rephrase.
|
line 49, revised -> Service workers face, communicate, and provide professional services in various person. (line 51-52) line 420 revised -> The job stress aggravated by providing their service in person in the COVID-19 era (line 428) |
|
Line 133 , and anywhere else the term occurs – still not sure what you mean by meaning of life – I think you mean quality of life. This is never well-defined in this paper. Because it is such an integral part of the argument, including in your statistical analysis, it must be well-defined or the paper doesn’t make sense. Even the explanation in 2.3.3 does not do much to explain and actually just ends up making the issue more confused.
|
Meaning in life is thought to be important to well-being throughout the human life span.(reference 29) For humans, having a clear idea of what to live for in their lives, that is, finding the purpose of their existence, is very important and indispensable (Frankl, 1959; Frankl, 1992). It can be said that it is a human's innate inner desire to find the meaning of their existence in life (Steger, Kashdan, Sullivan & Lorentz, 2008). The meaning in life is composed of sub-areas of meaning pursuit and meaning discovery. The meaning of life functions as an adaptive function to buffer the effects of negative events on psychological stress It revealed the importance of the meaning of life as a therapeutic and preventive resource that can increase an individual's well-being by coping with stress(ref.40). The pursuit of meaning can be defined as referring to the orientation or motivation to find meaning, and the discovery of meaning can be defined as referring to the subjective feeling that there is actually meaning in one's life.(ref. 18) Therefore, the meaning in life used in this study was judged to be able to buffer the stress of women bankers during the Covid-19, so it was considered as a research variable and tried to identify the factors that affect it by dividing the sub-areas. The overall meaning in life was found to be an influence factor. In addition, there is a confidence that the more difficult a situation is, the more difficult it is to find the root and try not to be shaken, the better the psychological well-being. Psychologically, the meaning in life is a major variable.
Therefore, it has a different meaning from the meaning in life and quality of life, and as the reviewer said, I will try to make it a study without confusion in the next study. Please understand. |
|
Line 149 – why remove people with less work experience? Why this threshold?
|
According to a survey and study in Korea, the work adaptation period of female office workers was 5.7 months. https://biz.chosun.com/site/data/html_dir/2007/08/06/2007080600456.html
Therefore, the subjects of this study were selected after the adaptation period for research related to understanding their duties and performing their duties as women bankers. |
|
Line separation on table 2 is confusing. |
Table 2 has been edited for readability. |

This manuscript is a resubmission of an earlier submission. The following is a list of the peer review reports and author responses from that submission.
Round 1
Reviewer 1 Report
The article has a very important and up-to-date subject, the well-being of women working in customer services. The article is well written and the conclusions are consistent with the results. I compliment the authors on their work and offer some suggestions for improvement to follow.
The methodological approach is standard, it does not present an original approach. Also, in the materials and methods section, it is important to reference the authors who recommend the validity parameters of the sampling and methodology used.
The chosen population/sample adds new perspectives on the effect of the pandemic. Thus, it is important to present more information about the women, for example, their nationality or country of residence. Also, I recommend clarifying how the difference in scales (7 points, 5 points, 3 points, and 4 points) was addressed in the analysis of the results.
I strongly recommend presenting the theoretical and practical implications in a specific section.
Also, in the conclusion, I strongly recommend identifying the limitations of the study and suggestions for future research.
Author Response
Thank you very much for your careful review so that it can be a good paper. As you pointed out, I worked hard to revise it. Thank you so much. .
|
Points to note |
revise |
|
The methodological approach is standard, it does not present an original approach. Also, in the materials and methods section, it is important to reference the authors who recommend the validity parameters of the sampling and methodology used. |
I modified it a lot to increase originality in the introduction part. The validity of this study was supported by adding previous studies that fit the validity and methodology of sampling. |
|
- The chosen population/sample adds new perspectives on the effect of the pandemic. Thus, it is important to present more information about the women, for example, their nationality or country of residence. -Also, I recommend clarifying how the difference in scales (7 points, 5 points, 3 points, and 4 points) was addressed in the analysis of the results. |
I added the situation of women by country(USA, China). -In particular, after COVID-19, in Korean society, the burden of care was further expanded to women, increasing the stress of working women and lowering job satisfaction -According to the statistician, it doesn't matter if all the measures used are not the same. |
|
I strongly recommend presenting the theoretical and practical implications in a specific section.
|
Theoretical and practical implications were added to the conclusion by dividing them into theoretical, research, and practical implications. |
|
Also, in the conclusion, I strongly recommend identifying the limitations of the study and suggestions for future research. |
The limitations and suggestions of the study were revised later in the discussion. Finally, it is suggested that the results of this study are used as follows in terms of nursing research, education, and practice. In terms of research, in order to maintain psychological well-being due to the prolonged Covid-19, it can be used as basic data for ultimately enhancing psychological well-being by classifying job-related stress for women in customer-facing service jobs and subdividing coping strategies to solve it. In terms of education, industry leaders can use it as a basis for education to increase the psychological well-being of working women responding to customers in the prolonged situation of Covid-19. At the practical level, it is necessary to subdivide and investigate the meaning of life, job stress, coping strategy, and resilience of working women in the service industry, and to develop and apply interventions to promote psychological well-being based on the community. However, in this study, instead of recruiting women working in various service fields, convenience sampling was performed in only some areas, so attention should be paid to generalizing the research results. Therefore, in preparation for unexpected situations such as Covid 19, it is proposed to diversify the selection of regions, industries, and subjects. |

Reviewer 2 Report
This paper analyzed factors related to the mental health of female service workers due to COVID-19.
This paper needs improvement in the following points.
1. Visualization of the study design process is required.
2. It is necessary to clearly organize the format of [Table 1].
In addition, it is necessary to clearly classify the contents of the table as age (year) < 30 20 (14.8).
Totals for all subjects are missing and must be displayed.
3. Each table should be captioned with a description of the variable or additional explanation for analysis.
4. Multiple regression analysis in Table 4 would have been considered according to the results in Table 3, but it seems that the subregions of each variable were not reflected.
5. According to the research results of this paper, it is judged that it is unreasonable to draw such discussion and conclusions. This is because the research results to support the discussion and conclusions are weak. Also, discussion is too verbose.
6. Lastly, there are many studies with very similar contents, no matter how recent the COVID-19 outbreak. This paper is not having Originality.
Author Response
Thank you very much for your careful review so that it can be a good paper. As you pointed out, I worked hard to revise it. Thank you so much. .
|
Points to note |
revise |
|
Visualization of the study design process is required. |
I added research design rather than research design process. |
|
2. It is necessary to clearly organize the format of [Table 1]. In addition, it is necessary to clearly classify the contents of the table as age (year) < 30 20 (14.8). Totals for all subjects are missing and must be displayed. |
Table 1 was modified to fit the format and reduced to important content.
The contents of the table have also been modified to be clearly categorized. |
|
3. Each table should be captioned with a description of the variable or additional explanation for analysis. |
I added an incidental explanation to each table. |
|
4. Multiple regression analysis in Table 4 would have been considered according to the results in Table 3, but it seems that the subregions of each variable were not reflected. |
Regression analysis was performed again with the results in Table 3, and revised it.
|
|
5. According to the research results of this paper, it is judged that it is unreasonable to draw such discussion and conclusions. This is because the research results to support the discussion and conclusions are weak. Also, discussion is too verbose. |
I revised the discussion and conclusion. I cut down on the discussion. Thank you. |
|
6. Lastly, there are many studies with very similar contents, no matter how recent the COVID-19 outbreak. This paper is not having Originality. |
According to the situation in Korea, there are not many studies on the well-being of service workers during the COVID-19 period. Since the situation in each country is different, I think research on this is significant. ã…‘worked hard to revise the contents to enhance originality.
In neighboring countries such as China and Japan, as the contagiousness increased due to COVID-19, patients and deaths continued, and the quality of life of the people, including women, declined by strengthening lockdown and social distancing with strong sanctions. In the case of Korea, the worst situation did not occur due to the initial response to the outbreak of infectious diseases and the active cooperation of the people, but the health and quality of life of the people were worse than before COVID-19.
|

Reviewer 3 Report
I have read with interest this paper. I believe that the paper is interesting. However, I have some concerns that are reported herein.
Introduction: I recommend to include the term “Quality of Working Life".
Variables: Authors should define all outcomes, exposures, predictors, potential confounders, and effect modifiers.
The variable "religion" should be discuss; Meaning of Life, Job Stress, Coping Strategy, and Resilience on Psychological Well-Being could be influenced by religion.
Bias and study size: authors should describe any efforts to address potential sources of bias and to explain how the study size was arrived at.
Stadistical methods: authors should explain how missing data were addressed.
Limitations: authos should discuss limitations of the study, taking into account sources of potential bias or imprecision. Discuss both direction and magnitude of any potential bias
Author Response
Thank you very much for your careful review so that it can be a good paper. As you pointed out, I worked hard to revise it. Thank you so much. .
|
심사3 |
|
|
Introduction: I recommend to include the term “Quality of Working Life". |
In introduction: I added the term “Quality of Working Life". I found the article and added the contents. Thank you. |
|
Variables: Authors should define all outcomes, exposures, predictors, potential confounders, and effect modifiers. |
Added definition of variable. The meaning of life, job stress, three coping strategies, and resilience were described as predictors, and religion was described as potential confounding variables. |
|
The variable "religion" should be discuss; Meaning of Life, Job Stress, Coping Strategy, and Resilience on Psychological Well-Being could be influenced by religion.
|
I mentioned religion. |
|
Bias and study size: authors should describe any efforts to address potential sources of bias and to explain how the study size was arrived at. |
Preliminary research was presented as a reference to determine the size of the study and precautions for bias in participants and procedure. Branch managers explained about the research at an executive meeting of other banking departments, and the branch executives who accepted it took questionnaires as many as female bankers. First of all, before going to the meeting, the researcher and research assistant had a meeting with the branch manager. Researchers trained and asked questions to familiarize themselves with the purpose of the study, how to collect data, and the contents of the questionnaire through two meetings in the VIP conference room of the banks. Executives at each branch took questionnaires by checking the number of research participants through meetings. The trained branch managers went to each bank's executive meeting and explained it to the executives who participated on behalf of the researchers. The trained branch managers went to each bank's executive meeting and explained it to the executives who participated on behalf of the researchers. The questionnaire was taken as many as the number of bank clerks at each branch, distributed to female bank clerks, and agreed to the written consent form. After that, the questionnaire was filled out and put in an envelope to come to the trained branch manager through the bank delivery network. When filling out the questionnaire, call the researcher's contact information if you have any questions. |
|
Stadistical methods: authors should explain how missing data were addressed. |
A total of 8 of the collected questionnaires that were omitted or responded faithfully were not included in the study, which was described in the subject's description. And the remaining 135 questionnaires used in the analysis were not missed. |
|
Limitations: authos should discuss limitations of the study, taking into account sources of potential bias or imprecision. Discuss both direction and magnitude of any potential bias |
I added a description as below. Finally, it is suggested that the results of this study are used as follows in terms of nursing research, education, and practice. In terms of research, in order to maintain psychological well-being due to the prolonged COVID-19, it can be used as basic data for ultimately enhancing psychological well-being by classifying job-related stress for women in customer-facing service jobs and subdividing coping strategies to solve it. In terms of education, industry leaders can use it as a basis for education to increase the psychological well-being of working women responding to customers in the prolonged situation of COVID-19. At the practical level, it is necessary to subdivide and investigate the meaning of life, job stress, coping strategy, and resilience of working women in the service industry, and to develop and apply interventions to promote psychological well-being based on the community. However, in this study, instead of recruiting women working in various service fields, convenience sampling was performed in only some areas, so attention should be paid to generalizing the research results. Therefore, in preparation for unexpected situations such as COVID-19, it is proposed to diversify the selection of regions, industries, and subjects. It is also suggested that religion be careful about interpretation and conduct research on this in the future because it can affect the variables of measuring of life, job Stress, coping strategy, and resilience on psychological well-being. |

Reviewer 4 Report
In the COVID-19 Era, Effects of Meaning of Life, Job Stress, Coping Strategy, and Resilience on Psychological Well-Being 3 of Women who work in Customer Services
Comments to Authors
The current study investigates the effects of the meaning of life, job stress, coping strategy, and resilience on psychological well-being in the COVID-19 era in a sample of women working in Customer Services. The paper provides sufficient scientific contribution to the panorama of studies investigating the impact of Covid 19 pandemic on population’s wellbeing, particularly women's difficulty in balancing work and care responsibilities during home confinement. The topic is innovative enough and qualitatively good to be published. However, the Authors should make some changes and explanations to improve the manuscript's quality for his acceptance.
In the ABSTRACT SECTION, the Authors reported the aim of their study inaccurately. Could the authors clarify or improve this? In addition, the Authors completely omitted information on study recruitment and setting and methods of administering questionnaires and / or interviews (i.e., was it an online survey or not, use of specific modules...). From row 26 to row 47 the Authors make a redundant premise that could be reduced or synthesized. In the INTRODUCTION SECTION the bibliography is not appropriate and updated. In general, the introduction is very little focused on the impact of COVID-19. There is scarce international literature on the effects of COVID-19 on the general population, specifically women, in terms of mental health. According to many international studies, it is well known that the COVID-19 pandemic has led to high levels of psychological distress, depression, and anxiety. Many studies focused on the effects of the pandemic on women’s mental health. Women have a higher prevalence of risk factors known to intensify during a pandemic, including chronic environmental strain, preexisting depressive and anxiety disorders, and domestic violence. Results from a cross-sectional survey of European working women showed that women working from home had a higher prevalence of symptoms of depression compared to those traveling to a job, suggesting that maintaining contact with people face-to-face was a significant protective factor against experiencing symptoms of depression during a period of social distancing. However, it would be helpful to mention some studies listed below. Rows 137-151. The description of the aims of the study need to be improved. In the manuscript, the term ‘female’ (see methods and discussion sections) would be better replaced by ‘woman’ (i.e. row 170). About the description of socio-demographic data, the table is rather exhaustive. Therefore the narrative description in the text of these same data appears redundant. Row 543. In the DISCUSSION SECTION about the following sentences “… people experience the growth that changes the hardship to something positive by perceiving pains 543 and introspecting themselves with their resilience..” are the Authors referring to the post-traumatic growth construct? If so, it would be better to clarify and describe the construct.The text could benefit from English language editing.
Suggested References
https://pubmed.ncbi.nlm.nih.gov/32215365/
https://pubmed.ncbi.nlm.nih.gov/33384623/
https://pubmed.ncbi.nlm.nih.gov/32155789/
https://pubmed.ncbi.nlm.nih.gov/32199182/
https://pubmed.ncbi.nlm.nih.gov/35463190/
https://www.ncbi.nlm.nih.gov/pmc/articles/PMC7152912/
https://pubmed.ncbi.nlm.nih.gov/33263142/
https://pubmed.ncbi.nlm.nih.gov/28458716/
Author Response
Thank you very much for your careful review so that it can be a good paper. As you pointed out, I worked hard to revise it. Thank you so much. .
|
심사4 |
|
|
Comments to Authors However, the Authors should make some changes and explanations to improve the manuscript's quality for his acceptance. |
Thank you. To improve the quality of my thesis, I made a lot of corrections while adding the papers you sent me.
|
|
In the ABSTRACT SECTION, the Authors reported the aim of their study inaccurately. Could the authors clarify or improve this? In addition, the Authors completely omitted information on study recruitment and setting and methods of administering questionnaires and / or interviews (i.e., was it an online survey or not, use of specific modules...).
|
Revised abstract as below. The purpose of this study was to identify the degree of psychological well-being and analyze the factors affecting psychological well-being by using variables of meaning of life, job stress, coping strategies, and resilience to improve the quality of working life during COVID-19. The participants were 135 adult women working for customers in banks. Data were collected using a self-written questionnaire and analyzed using t-test, ANOVA, Pearson’s correlation coefficients and multiple regression etc. The participants’ psychological well-being and meaning of life (r=.43, p<.001), resilience (r=.36, p<.001), the pursuit of social support (r=.33, p<.001), and problem-solving-focused coping (r=.55, p<.001) showed positive correlations. Psychological well-being and job stress (r=-.44, p<.001), and avoidance-focused coping (r=-.28, p=.001) showed negative correlations. The factors affecting psychological well-being were the problem-solving-focused approach (β=.35, p<.001), job role stress (β=-.24, p<.001), meaning of life (β=.29, p<.001), avoidance-focused coping (β =-.23, p<.001) and resilience (β=.15, p=.023). It is necessary to formalize psychological interventions to induce improvement of the quality of work life by increasing the psychological well-being of working women during COVID-19. It is required to develop a psychological well-being program in consideration of the problem-solving-oriented coping, job stress, meaning of life, avoidance-oriented coping, and resilience of working women. |
|
From row 26 to row 47 the Authors make a redundant premise that could be reduced or synthesized. In the INTRODUCTION SECTION the bibliography is not appropriate and updated. In general, the introduction is very little focused on the impact of COVID-19. There is scarce international literature on the effects of COVID-19 on the general population, specifically women, in terms of mental health. According to many international studies, it is well known that the COVID-19 pandemic has led to high levels of psychological distress, depression, and anxiety. Many studies focused on the effects of the pandemic on women’s mental health. Women have a higher prevalence of risk factors known to intensify during a pandemic, including chronic environmental strain, preexisting depressive and anxiety disorders, and domestic violence. Results from a cross-sectional survey of European working women showed that women working from home had a higher prevalence of symptoms of depression compared to those traveling to a job, suggesting that maintaining contact with people face-to-face was a significant protective factor against experiencing symptoms of depression during a period of social distancing. However, it would be helpful to mention some studies listed below. |
The introduction has been drastically revised. I also included the references you gave me. I focused on COVID-19 and described it from the beginning. . |
|
Rows 137-151. The description of the aims of the study need to be improved. In the manuscript, the term ‘female’ (see methods and discussion sections) would be better replaced by ‘woman’ (i.e. row 170). About the description of socio-demographic data, the table is rather exhaustive. Therefore the narrative description in the text of these same data appears redundant. |
I modified the purpose of the study.
. |
|
Row 543. In the DISCUSSION SECTION about the following sentences “… people experience the growth that changes the hardship to something positive by perceiving pains 543 and introspecting themselves with their resilience..” are the Authors referring to the post-traumatic growth construct? If so, it would be better to clarify and describe the construct. The text could benefit from English language editing.
|
I Modified as below. Thank you. When an individual experiences an event shock, he or she also experiences growth that changes difficulties into positive ones by recognizing the meaning of life, using coping strategies about stress well, and reflecting on himself or herself by increasing resilience |
Round 2
Reviewer 3 Report
The manuscript has been improved.
Reviewer 4 Report
The study still presents too many severe methodological problems.
The ABSTRACT was not improved, as suggested (questionnaire self-written???)
The INTRODUCTION is excessively long and difficult to read.
row 157-159 ... "this study understands!!!!! ..."
MATERIALS AND METHODS section
The paragraph Procedures and Measures revisions are not clear.
The RESULTS section has not been improved.
We still observe the systematic error in reporting the figures in the RESULTS and the DISCUSSION section. The DISCUSSION section had to be made shorter and more incisive.
The suggestions provided were intended to lead to a significant improvement, which we do not observe.
The Bibliography is almost entirely centered on Korean works, and comparing the results with an international panorama isn't straightforward.
In our opinion, English needs to be much improved.